# Preimplantation Testing for Polygenic Disease (PGT-P): Brave New World or Mad Pursuit?

**Darren K. Griffin [1],\* and Anthony T. Gordon [2]**

[1]  School of Biosciences, University of Kent, Canterbury CT2 7NJ, UK
[2]  Cooper Surgical, 84 Wood Lane, London W12 0BZ, UK
\*   Correspondence: d.k.griffin@kent.ac.uk

**Abstract:** In preimplantation testing for monogenic disease (PGT-M), we are used to specific and directed diagnoses. Preimplantation testing for polygenic disease (PGT-P), however, represents a further level of complexity in that multiple genes are tested for with an associated polygenic risk score (PRS), usually established by a genome-wide association study (GWAS). PGT-P has a series of pros and cons and, like many areas of genetics in reproductive medicine, there are vocal proponents and opponents on both sides. As with all things, the question needs to be asked, how much benefit does PGT-P provide in comparison to the risks involved? For each disease, a case will need to be made for PGT-P, as will a justification that the family involved will actually benefit; the worry is that this might be more work than the cost justifies.

**Keywords:** PGT-P; polygenic risk score; genome wide association study

## 1. Background

In preimplantation testing for monogenic disease (PGT-M), we are used to specific and directed diagnoses. For instance, we can tell with reasonable accuracy whether an embryo will lead to a live birth with cystic fibrosis or Tay Sachs disease by virtue of the fact that it carries two mutant recessive genes (alleles). By and large, these recessive traits have similar symptoms from person to person, i.e., genotype to phenotype correlations are relatively straightforward, albeit needing to be confirmed by prenatal diagnosis in case of false negative results. With dominant traits such as Hypercholesterolemia, Tuberous Sclerosis, or Neurofibromatosis, the situation is more complex as only a single copy of the disease gene is required (usually present already in one parent) but the severity of the disease can vary greatly due to variable expressivity. Further complications again occur when not only a single gene disorder is screened for (such as Fanconi's anaemia) but, at the same time, because we also need to ask whether an embryo is a tissue match as a donor for a pre-existing affected child (HLA-typing—so-called "saviour siblings"). Essentially, the situation gets more complicated each time another trait is screened for. Preimplantation testing for polygenic disease (PGT-P) represents a further level of complexity.

Around 2–5% of live births suffer congenital disorders, the majority of which are thought to be monogenic [1], although each are individually rare. In point of fact, the diseases of most worry globally are those with a partial, complex genetic component, such as cancer, schizophrenia, diabetes and heart disease. The genetic basis of these common disorders is considered polygenic; that is, in the pathology of these diseases more than one gene being involved. Environmental factors also may also play a role, in which case they are considered "multifactorial". For instance, diet, radiation exposure, smoking, exercise, etc. can all influence the risk of contracting cancer. We all carry a risk of each of these and other similar diseases; the risk of contracting cancer in our lifetime, for instance, is currently quoted as 50% [2]. The overall risk of these disorders can be assessed by calculating a probability known as a polygenic risk score, taking environmental factors into account.

## 2. Polygenic Risk Scores

The term "PRS" (polygenic risk score) is used to estimate a person's risk of developing a particular disease based on their genetics. Assigning a PRS to an embryo is possible because many groups have studied large datasets of people with each disease. PRSs are, in turn, derived from genome-wide association studies (GWAS).

GWAS studies take the DNA of multiple individuals, annotated with outcomes from polygenic diseases, applying each DNA individually to a "SNP chip" (single nucleotide polymorphism microarray). More recently, genome-wide whole genome and exome sequencing has superseded SNP chip data. Variations in DNA sequence (SNPs) differ from individual to individual, and for each locus, an individual person is assigned "AA", "AB", or "BB", which is determined by interpreting the fluorescent signals on the chip. These have been performed for schizophrenia, type 1 and 2 diabetes, heart disease/attack, Alzheimer disease, breast cancer, prostate cancer, testicular cancer, basal cell carcinoma, malignant melanoma, hypercholesterolemia, hypertension, heart attack, coronary artery disease, atrial fibrillation, stroke, and inflammatory bowel disease, amongst others.

This array of AA/AB/BB data provides the raw material for the PGT-P risk analysis algorithm. Therein, the embryo biopsy is taken, the DNA is amplified, and then applied to the SNP chip or analysed by sequencing (whole genome or exome). By interpreting the minor and major alleles at the relevant loci, the PRS is generated for each embryo for each disease [3]. The PRS model assumes that the embryo has a similar ethnic background to the study population in the original GWAS and that it will undergo similar lifestyle choices [3]. Unlike PGT-M, then, the result is probabilistic, not predictive. Moreover, the fact that many groups are studying or have studied PRSs does not necessarily mean that these can be applied easily to embryos. There are already ongoing issues with their application in postnatal life, and a large amount of research is being undertaken that will only be resolved with sound cohort longitudinal studies. Even then, the impact of PRSs in liveborn individuals, most commonly adults, compared to prenatally, is likely to be different due to different environmental exposures.

The PGT assessment for predisposition disorders already has a history, albeit for monogenic disorders with a known risk score. It has its roots in the Delhanty lab, where a cancer predisposition syndrome APC PGT case was first reported [4]. There are also examples of PGT being performed for gene variants that carry a calculable risk of breast cancer, such as those in the BRCA1 and BRCA2 genes [5]. Treff et al. argue that an improved diagnosis could be performed using PRSs from several genes, including BRCA1 and BRCA2 in a PGT-P strategy, that near guarantees (~100%) the development of breast cancer [6]. They argue that this is one of the prime examples why PGT-P is justified [6].

PRSs are now being used in adult populations to advise people of lifestyle choices, effects of certain medicines, etc., but the crux of the question here is their application on embryos. Embryo screening remains a very emotive subject and, in the past, geneticists have been our own worst enemies in giving a false impression that our genome (indeed, genetics) is predictive and pre-deterministic when, in fact, it rarely is.

## 3. Positives in Favour of PGT-P

Some people would argue that, in any PGT cycle with embryos deemed suitable for transfer, you have to rank the embryos anyway, so adding a PRS is only another means of facilitating selection. An embryo could be screened for aneuploidy (PGT-A), for morphology (both indicators have, sometimes controversially, been associated with the success of establishing a pregnancy), for the presence of a hitherto unlooked for serious genetic disease, and, finally, for PRS pertaining to several conditions. Using computer algorithms that take into account patients' preferences and concerns (established by a questionnaire) and then a "rank-order" for each embryo to be implanted could be achieved. This should not be hard in principle.

A second argument is that, for some diseases (e.g., dominant disorders) we already do not know the penetrance. That is, even if an embryo has the dominant disease gene,

will the subsequent baby/child/adult get the disease, and with what severity if so? By extension, therefore, PGT-P is not such a big next step.

Third, another argument is that if PGT-P were practiced widely, it might relieve the burden on healthcare systems, as people with a lower PRS (e.g., heart disease or cancer patients) would result from the process.

Fourth, purportedly for specific disorders, PGT-P is a significant improvement on the current PGT-M approach (see above for a caveat) [6]. That is, adding more genes to the panel (BRCA1 and 2) can more accurately predict breast cancer risk, it has been argued [6].

Finally, the UK has been a pioneer of this type of genomic medicine (including the 100 K genomes public database). It gave us IVF, PGT, karyomapping, saviour siblings, and nuclear genome transfer for mitochondrial disease. This was achieved in part through groundbreaking science and medicine, but also through a robust and sensible ethico-legal framework. As a result, many previously controversial issues (PGT-M for instance) are now considered routine and socially acceptable. Adopting PGT-P under strict HFEA regulation and on a case-by-case basis should be no different.

## 4. Risks of PGT-P

- Risk without benefit: This treatment could lead to putting patients at risk of the complications associated with IVF (particularly in young, good prognosis patients) with only minimal demonstrable benefits afforded by PGT-P.
- Available only for the rich: The thorny issue to all of this type of treatment is that it is not free at the point of delivery. Richer families are therefore inevitably more likely to benefit from it, and the process could lead to exploitation of families desperate to have children, exposing them to another "add-on" with no proven clinical benefit.
- Slippery slope: By introducing PGT-P, the opportunity for using it for more cosmetic reasons (e.g., height, IQ) less clearly associated with health becomes more likely. Moreover, could the licensing of PGT-P be seen as a prelude to the licensing of the more controversial gene editing or manipulation of embryos? Some say that this is inevitable.
- Encouraging complacency: If a person or parents are safe in the knowledge that they/their child are in a genetically lower risk group of cancer or heart disease, to take two examples, would they be less likely to follow a more healthy lifestyle, thereby negating the benefits of the PGT-P?
- How do we select the families? Which families should be selected for PGT-P, and why? This is not a straightforward question, and it could depend on a series of factors that are not necessarily related to clinical need. The data is based on the 100,000 genomes database, which is 85% Caucasian—other racial backgrounds (including admixtures) might not know whether they should be in a high-risk group.
- Incidental findings and pleiotropic effects: There are many genetic variants for which we do not know (or only partially know) the clinical significance. Many of these will be revealed (potentially) through PGT-P, without proper knowledge or evaluation. These traits do not occur in a vacuum; by selecting against the likelihood of a certain disease, we may be selecting against further traits. Moreover, PGT-P are selected for one polygenic phenotype but since we don't fully understand the inter-relationships between polygenic trait genes we could accidentally be selecting for another undesirable phenotype?
- Effects of rare SNPs: Some SNPs that are under-represented in databases may have high impact on some PRS and thus skew the interpretation of results.
- Public backlash: We know through experience of GM crops, to take an example, that the public can react badly to certain technologies. Could PGT providers be damaged financially and reputationally as a result of offering PGT-P putting the whole field in jeopardy? People are acutely aware of the "eugenics" terminology and the press for PGT-P thus far has not been good.

- Too much information: Provided with incredible amounts of genomic information that even the experts find it difficult to interpret, there is a genuine risk of obsessive over-interpretation of the PRS on the part of the patients. This could place an unreasonable burden on genetic counsellors—relative risk is hard enough to establish, even at the best of times.
- Times change: The datasets are based on current people's lifestyles. These will inevitably change by the time the disease develops (or not) in the current embryo. There may also be more effective treatments for these diseases by the time that they become an issue. This applies to all late onset diseases, whether they be monogenic or polygenic.
- Evidence-based medicine: Although it will be possible to ascertain that the foetus has the same genomic sequence as the original diagnosis in the embryo, it really will not be practicable to establish, for most polygenic disorders, whether PGT-P has significantly reduced the chances of the diseases developing. The follow-up of cases could take decades. Some children will also get the disease for which they were diagnosed with a low PRS. Families might not hear the message, "we only told you that your risk was lower".

## 5. Cost-Benefit Ratio

As with all things, the question needs to be asked, how much benefit are you actually creating in comparison to the risks involved? In order to establish this as a service, going forward, the community will need to invest significant time and effort. In our personal opinion, there may be a limited number of conditions and a limited number of families for which, under strict regulatory oversight, PGT-P may be appropriate. Indeed, some authors make a very convincing case that the BRCA1 and 2 PGT-P approach is more accurate than the current PGT-M version [6]. For each disease, a case will need to be made as will a justification that the family involved will actually benefit. This might be more work than the cost justifies.

Another key consideration is whether PGT-P could or would be used simply to rank embryos, or would it be used as a justification to "not transfer" a human embryo. The difference is significant, as, ethically speaking, the latter could lead to destroying high risk embryos.

When PGT first started back in the late 1980s, we were warned that this day would come. In the intervening period, we have always argued against polygenic screening because of the risks involved—but times change. This is an interesting medical, technical and ethical challenge in the light of new technology and the historical success of PGT. The main things that practitioners have to consider are regulation (and how they will react—they are very invested in evidence-based medicine at the moment and "add-ons" are not viewed favourably). To what extent can clinics claim to be reducing the risk of disease when you have little evidence that you are so doing? A final point in the cost-benefit analysis is how "pointless" this is? When we have eliminated the aneuploids, the grossly abnormal embryos, in a regular PGT cycle, then how many will be left to look at PRS and how much will each embryo be different from the next? What mad pursuit will ensue to in a situation where we are agonising over 55% vs. 50% PRS compared to subtle differences in morphological characteristics?

**Author Contributions:** Conceptualization, D.K.G. and A.T.G.; methodology, D.K.G. and A.T.G.; writing—original draft preparation, D.K.G.; writing—review and editing, A.T.G. All authors have read and agreed to the published version of the manuscript.

**Funding:** This research received no external funding.

**Institutional Review Board Statement:** Not applicable.

**Informed Consent Statement:** Not applicable.

**Data Availability Statement:** There is no further data to this manuscript.

**Conflicts of Interest:** Tony Gordon is an employee of Cooper Surgical who could, theoretically, benefit financially from publication of this article.

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
