# Peer review of "Preimplantation Testing for Polygenic Disease (PGT-P): Brave New World or Mad Pursuit?"

_2673-8856, doi:10.3390/dna3020008_

Round 1

Reviewer 1 Report

Preimplantation testing for polygenic disease (PGT-P): Brave new world or ad pursuit?

This is an interesting and very didactive paper, dealing with the Pros and the Cons of PGT-P. The subject is a matter of debate among the geneticist community, because the knowledge on PRS remains scarce and the potential benefit of using such scores to rank human embryos not obvious.

I have a quite few remarks listed below.

Major points

First of all, it is not clear if PGT-P would be used to rank embryos, or would they use it to “not transfer” a human embryo. The difference is noticeable, because ethically speaking, it is completely different to favor the transfer of low risk embryos or to destroy high risk embryos. The authors should have a comment of this.

In the Introduction section, second paragraph, the definition of polygenic seems not correct to me. A polygenic disorder is a disorder influenced by 2 or more genes (ex: eye color). Many polygenic traits are influenced by the environment and then are called “multifactorial diseases”.

Minor points

At the end of the abstract there is 2 dots.

In the Introduction section, first paragraph, page 1 lines 24-27 the authors should mention the terms of “variable expressivity” when they explain that “the severity of the disease can vary greatly”. Lines 28 delete the inappropriate “a”.

In the “polygenic risk scores” section, first sentence, I would rather say that the polygenic score is used to estimate a person's risk for a particular disease based on their genetics. Please delete the second “The”.  

Page 2, lines 57, 58, when the authors mention “this provides the data for the PGT-P risk” what does “this” stand for?

In the “positives in favour of PGT-P” section, first sentence, please delete “This”.  

Is PGT-A a real indicator of the success of establishing a pregnancy (line 80)? The authors should rephrase this point because it remains debated from my point of view.

Could the authors better explain the fourth point about the improvement of the PGT-M approach? It is not clear for me. Do the authors mean that PGT-P can be added to a more classical PGT-M to increase the “cancer-risk reduction”?

In the “Risks of PGT-P” section, the point about risk without benefit should appear first in the list, because this is the major problem from my point of view.

Please edit GM (line 132, page 4).

Author Response

Preimplantation testing for polygenic disease (PGT-P): Brave new world or ad pursuit?

This is an interesting and very didactive paper, dealing with the Pros and the Cons of PGT-P. The subject is a matter of debate among the geneticist community, because the knowledge on PRS remains scarce and the potential benefit of using such scores to rank human embryos not obvious. 

I have a quite few remarks listed below.

Major points

First of all, it is not clear if PGT-P would be used to rank embryos, or would they use it to “not transfer” a human embryo. The difference is noticeable, because ethically speaking, it is completely different to favor the transfer of low risk embryos or to destroy high risk embryos. The authors should have a comment of this.

Thank you for this comment – we have added a couple of sentences to this effect.

In the Introduction section, second paragraph, the definition of polygenic seems not correct to me. A polygenic disorder is a disorder influenced by 2 or more genes (ex: eye color). Many polygenic traits are influenced by the environment and then are called “multifactorial diseases”. 

Point taken – definition amended

Minor points

At the end of the abstract there is 2 dots.

Corrected

In the Introduction section, first paragraph, page 1 lines 24-27 the authors should mention the terms of “variable expressivity” when they explain that “the severity of the disease can vary greatly”. Lines 28 delete the inappropriate “a”.

 Done

In the “polygenic risk scores” section, first sentence, I would rather say that the polygenic score is used to estimate a person's risk for a particular disease based on their genetics. Please delete the second “The”.  

Done

Page 2, lines 57, 58, when the authors mention “this provides the data for the PGT-P risk” what does “this” stand for?

“This” has been clarified

In the “positives in favour of PGT-P” section, first sentence, please delete “This”.  

Done

Is PGT-A a real indicator of the success of establishing a pregnancy (line 80)? The authors should rephrase this point because it remains debated from my point of view. 

Rephrased as requested

Could the authors better explain the fourth point about the improvement of the PGT-M approach? It is not clear for me. Do the authors mean that PGT-P can be added to a more classical PGT-M to increase the “cancer-risk reduction”? 

An extra sentence of clarification added

In the “Risks of PGT-P” section, the point about risk without benefit should appear first in the list, because this is the major problem from my point of view. 

Done

Please edit GM (line 132, page 4).

Done

Reviewer 2 Report

The manuscript deals with a very "hot" topic. However, there are important flaws in the presentation of the scientific background and of the potential pros and cons of PGT-P. 

Just to make a few examples:

1. In the initial sentence, PGT-M is reported as a procedure that can lead to a definitive diagnosis. This statement is not accurate, since false negatives can occur, and the test result must be confirmed by invasive prenatal diagnosis. Perhaps the authors refer to the match between presence of a genetic variant causing a monogenic disorder and appearance of the phenotype, which is a different concept.

2. At the end of page 1, the statement that "The overall risk of these disorders being observed can be assessed by calculating a probability known as a polygenic risk score" is inaccurate. The risk is determined also by the environmental factors. Furthermore, there is some redundancy in the sentence (not sure why "being observed" is there?). 

3. The fact that many groups (by the way, there is no reference to the papers published by these groups) are studying or have studied PRSs does not imply that these can be applied easily to embryos. Indeed, there are still issues with their application in postnatal life, and a lot of research is going on. One will be able to know only with sound cohort longitudinal studies, since it is different to study PRSs in liveborn individuals, most commonly adults, rather than prenatally, when one does not know anything about environmental exposure. 

4. On page 2, the statement that "GWAS studies take the DNA of multiple individuals in public databases" is either incorrect or badly written. Cohorts of individuals enrolled for large studies are studied and tested, it's not clear why they should be "in public databases" and what this means. Some studies may allow the transfer of part of the data to public GWAS databases, but this are secondary products of the research.

There are several sentences that are very hard to understand (i.e., just as ana example, on page 2 "Indeed, there are some PRS ... breast cancer). Overall, the text contains many inaccuracies.

Author Response

The manuscript deals with a very "hot" topic. However, there are important flaws in the presentation of the scientific background and of the potential pros and cons of PGT-P. 

Just to make a few examples:

  1. In the initial sentence, PGT-M is reported as a procedure that can lead to a definitive diagnosis. This statement is not accurate, since false negatives can occur, and the test result must be confirmed by invasive prenatal diagnosis. Perhaps the authors refer to the match between presence of a genetic variant causing a monogenic disorder and appearance of the phenotype, which is a different concept.

We see the point of have adjusted the text accordingly. The point about definitive diagnoses was the fact that the PGT-M is meant to be definitive for the specific mutation. We have changed this to “specific and directed” to avoud any confusion that we are referring to genotype-phenotype correlations

  1. At the end of page 1, the statement that "The overall risk of these disorders being observed can be assessed by calculating a probability known as a polygenic risk score" is inaccurate. The risk is determined also by the environmental factors. Furthermore, there is some redundancy in the sentence (not sure why "being observed" is there?). 

We have removed “being observed” as requested and added a note about environmental factors.

  1. The fact that many groups (by the way, there is no reference to the papers published by these groups) are studying or have studied PRSs does not imply that these can be applied easily to embryos. Indeed, there are still issues with their application in postnatal life, and a lot of research is going on. One will be able to know only with sound cohort longitudinal studies, since it is different to study PRSs in liveborn individuals, most commonly adults, rather than prenatally, when one does not know anything about environmental exposure. 

We agree and feel that we too have made this point. As the reviewer has raised the issue however we have reinforced it with a few sentences.

  1. On page 2, the statement that "GWAS studies take the DNA of multiple individuals in public databases" is either incorrect or badly written. Cohorts of individuals enrolled for large studies are studied and tested, it's not clear why they should be "in public databases" and what this means. Some studies may allow the transfer of part of the data to public GWAS databases, but this are secondary products of the research.

We have rephrased the sentence

There are several sentences that are very hard to understand (i.e., just as ana example, on page 2 "Indeed, there are some PRS ... breast cancer). Overall, the text contains many inaccuracies.

We are happy to correct any “inaccuracies” that the reviewer perceives that we have. No more are listed in this review however. We have rephrased the one mentioned.

Reviewer 3 Report

The authors clearly presented the question of the polygenic disease testing by comparing it with the current PGT-M.  The opinion summarize pros and cons of the technique published in Nature protocols by Choi, S.W., et al. In chapter 2 it would be helpful if the author could better detail the “various cancer, diabetes…etc.” by listing them or by using a table in order to have a better view of the diseases involved in the screening.

In addition, would the author compare the present PGT-P with a direct exome sequencing performed on embryo’s biopsies?

Author Response

The authors clearly presented the question of the polygenic disease testing by comparing it with the current PGT-M.  The opinion summarize pros and cons of the technique published in Nature protocols by Choi, S.W., et al. In chapter 2 it would be helpful if the author could better detail the “various cancer, diabetes…etc.” by listing them or by using a table in order to have a better view of the diseases involved in the screening. 

An indicative list has been added. To be clear however, the point of the paper is to compare the whole practice of PGT-P not just a specific technique. The Choi paper is listed as an example of how PRS are calculated. It is not a methodology through which PGT-P is performed. As we have clearly caused some confusion to the reviewer, we have added a line of clarification in section 2.

In addition, would the author compare the present PGT-P with a direct exome sequencing performed on embryo’s biopsies?

Again, the purpose of the paper is not to compare methodologies, that is way beyond its scope. However we have added a line that PGT-P can be performed by SNP-chips, NGS or whole exome sequencing.

Round 2

Reviewer 2 Report

The authors have made some corrections, specifically concerning the few examples provided,  but, as mentioned in the previous review,  overall it has important flaws. The background is scanty and imprecise, very few references are used, and parts of the manuscript are not understandable. It should be re-written completely, using  proper scientific wording and w adequate references and improving the scientific contents.